# Connectivity map of bipolar cells and photoreceptors in the mouse retina

Christian Behrens[1,2,3,4†], Timm Schubert[1,2†], Silke Haverkamp[5], Thomas Euler[1,2,3], Philipp Berens[1,2,3]*

[1]Institute for Ophthalmic Research, University of Tübingen, Tübingen, Germany; [2]Center for Integrative Neuroscience, University of Tübingen, Tübingen, Germany; [3]Bernstein Center for Computational Neuroscience, University of Tübingen, Tübingen, Germany; [4]Graduate Training Center for Neuroscience, International Max Planck Research School, University of Tübingen, Tübingen, Germany; [5]Institute of Cellular and Molecular Anatomy, Goethe-University Frankfurt, Frankfurt, Germany

**Abstract** In the mouse retina, three different types of photoreceptors provide input to 14 bipolar cell (BC) types. Classically, most BC types are thought to contact all cones within their dendritic field; ON-BCs would contact cones exclusively via so-called invaginating synapses, while OFF-BCs would form basal synapses. By mining publically available electron microscopy data, we discovered interesting violations of these rules of outer retinal connectivity: ON-BC type X contacted only ~20% of the cones in its dendritic field and made mostly atypical non-invaginating contacts. Types 5T, 5O and 8 also contacted fewer cones than expected. In addition, we found that rod BCs received input from cones, providing anatomical evidence that rod and cone pathways are interconnected in both directions. This suggests that the organization of the outer plexiform layer is more complex than classically thought.

*For correspondence: philipp. berens@uni-tuebingen.de

†These authors contributed equally to this work

Competing interests: The authors declare that no competing interests exist.

## Introduction

Parallel visual processing already starts at the very first synapse of the visual system, where photoreceptors distribute the signal onto multiple types of bipolar cells. In the mouse retina, two types of cone photoreceptors differing in their spectral properties – short (S-) and medium wavelength-sensitive (M-) cones – and rod photoreceptors provide input to 14 types of bipolar cell (reviewed in *Euler et al., 2014*). The precise connectivity rules between photoreceptors and bipolar cell (BC) types determine which signals are available to downstream circuits. Therefore, the connectome of the outer retina is essential for a complete picture of visual processing in the retina.

For some mouse BC types, specific connectivity patterns have already been described: For example, based on electrical recordings and immunohistochemistry cone bipolar cell type 1 (CBC1) have been suggested to selectively contact M-cones, whereas CBC9 exclusively contacts S-cones (*Haverkamp et al., 2005*; *Breuninger et al., 2011*). The other BC types are thought to contact all M-cones within their dendritic field, but their connectivity to S-cones is unclear (*Wässle et al., 2009*). In addition, two fundamental cone-BC contact shapes have been described: invaginating contacts with the dendritic tips extending into the cone pedicle and flat (basal) contacts that touch the cone pedicle base, commonly associated with ON- and OFF-BCs, respectively (*Dowling and Boycott, 1966*; *Kolb, 1970*; *Hopkins and Boycott, 1995*).

Rod bipolar cells (RBCs) are thought to exclusively receive rod input and to feed this signal into the cone pathway via AII amacrine cells (reviewed by *Bloomfield and Dacheux, 2001*). However, physiological data indicate that RBCs may receive cone photoreceptor input as well (*Pang et al., 2010*). Also, types CBC3A, CBC3B and CBC4 have been reported to receive direct rod input

(*Mataruga et al., 2007*; *Haverkamp et al., 2008*; *Tsukamoto and Omi, 2014*), suggesting that rod and cone pathways are much more interconnected than their names implicate.

Here, we analyzed an existing electron microscopy dataset (*Helmstaedter et al., 2013*) to quantify the connectivity between photoreceptors and bipolar cells in the mouse. We did not find evidence for additional M- or S-cone selective CBC types in addition to the reported CBC1 and 9. However, we found interesting violations of established rules of outer retinal connectivity: The newly discovered CBCX (*Helmstaedter et al., 2013*), likely an ON-CBC (*Ichinose et al., 2014*), had unexpectedly few and mostly atypical basal contacts to cones. CBC5T, CBC5O and CBC8 also contacted fewer cones than expected from their dendritic field. In addition, we provide anatomical evidence that rod and cone pathways are connected in both directions: Not only OFF-types CBC3A, CBC3B and CBC4 get direct input from rods but also RBCs from cones.

## Results

### Identification of S- and M-cones

We used the serial block-face electron microscopy (SBEM) dataset *e2006* published by *Helmstaedter et al. (2013)* to analyze the connectivity between photoreceptors and bipolar cells in the outer plexiform layer (OPL) of the mouse retina (*Figure 1A*). To this end, we reconstructed the volume of all cone axon terminals (cone pedicles; n = 163) in the dataset as well as the dendritic trees of all BCs (n = 451; *Figure 1B*, see Materials and methods).

To identify S- and M-cones, we used the fact that type nine cone bipolar cells selectively target S-cones (*Figure 1C,D*) (*Mariani, 1984*; *Kouyama and Marshak, 1992*; *Haverkamp et al., 2005*; *Breuninger et al., 2011*). We found 48 contacts of CBC9s and cones, involving 43 cones (*Figure 1— figure supplement 1A*). We visually assessed all contacts and found that 29 of these were in the periphery of the cone pedicle, where no synapses are expected (*Figure 1—figure supplement 1B*) (*Dowling and Boycott, 1966*; *Chun et al., 1996*). This left 14 potential S-cones with invaginating contacts from at least one CBC9. It has been shown that S-cones are contacted by the dendrites of all neighboring CBC9s and that these contacts occur mostly at the tip of dendritic branches (*Haverkamp et al., 2005*). Out of the 14 candidate cones, eight cones had only one CBC9 contact. Some of these cones were contacted by a CBC9 dendritic branch that continued past the contact site. Other cones – although contact by one CBC9 – were not contacted by passing dendrites from other CBC9s. The other six cones had at least two invaginating contacts from CBC9s. These originated from two different CBC9s or – in case they originated only from a single CBC9 – at least one of them was formed by a dendritic branch ending at the cone (*Figure 1E*). We labeled the eight cones that featured only a single CBC9 contact as M-cones (*Figure 1—figure supplement 1C*), and defined the remaining six candidate cones as S-cones (*Figure 1D* and *Figure 1—figure supplement 1C and D*, see Materials and methods). This corresponds to a fraction of 4.8% S-cones (6/124 cones within the dendritic field of at least one CBC9), matching the 3–5% reported in previous studies (*Röhlich et al., 1994*; *Haverkamp et al., 2005*).

An alternative scheme for identifying S-cones would have been to classify all cones with at least one invaginating contact from CBC9 as S-cones. This would have resulted in a total of 14 S-cones out of 124 cones (*Figure 3—figure supplement 2A*) or a fraction of 11.3%. Because this S-cone percentage is much larger than the 3–5% reported earlier (*Haverkamp et al., 2005*), we consider this scenario as very unlikely (p=0.0037, binomial test, null hypothesis: 5% S-cones, n = 124).

### Classification of photoreceptor-BC contacts

We next developed an automatic method to distinguish contacts likely corresponding to synaptic connections from false contacts. As the tissue in the dataset was stained to enhance cell-surface contrast to enable automatic reconstruction, it is not possible to distinguish between synaptic contacts based on explicit ultrastructural synaptic markers, such as vesicles, synaptic ribbons or postsynaptic densities (see also discussion in *Helmstaedter et al., 2013*). In contrast to the synaptic contacts in the inner plexiform layer studied by Helmstaedter et al. (*Helmstaedter et al., 2013*), the highly stereotypical morphology of synapses at photoreceptor axon terminals allowed us to classify the contacts (*Haverkamp et al., 2000*): The ribbon synapses of the cones are placed exclusively in the presynaptic area at the bottom of the cone pedicles. Here, ON-cone

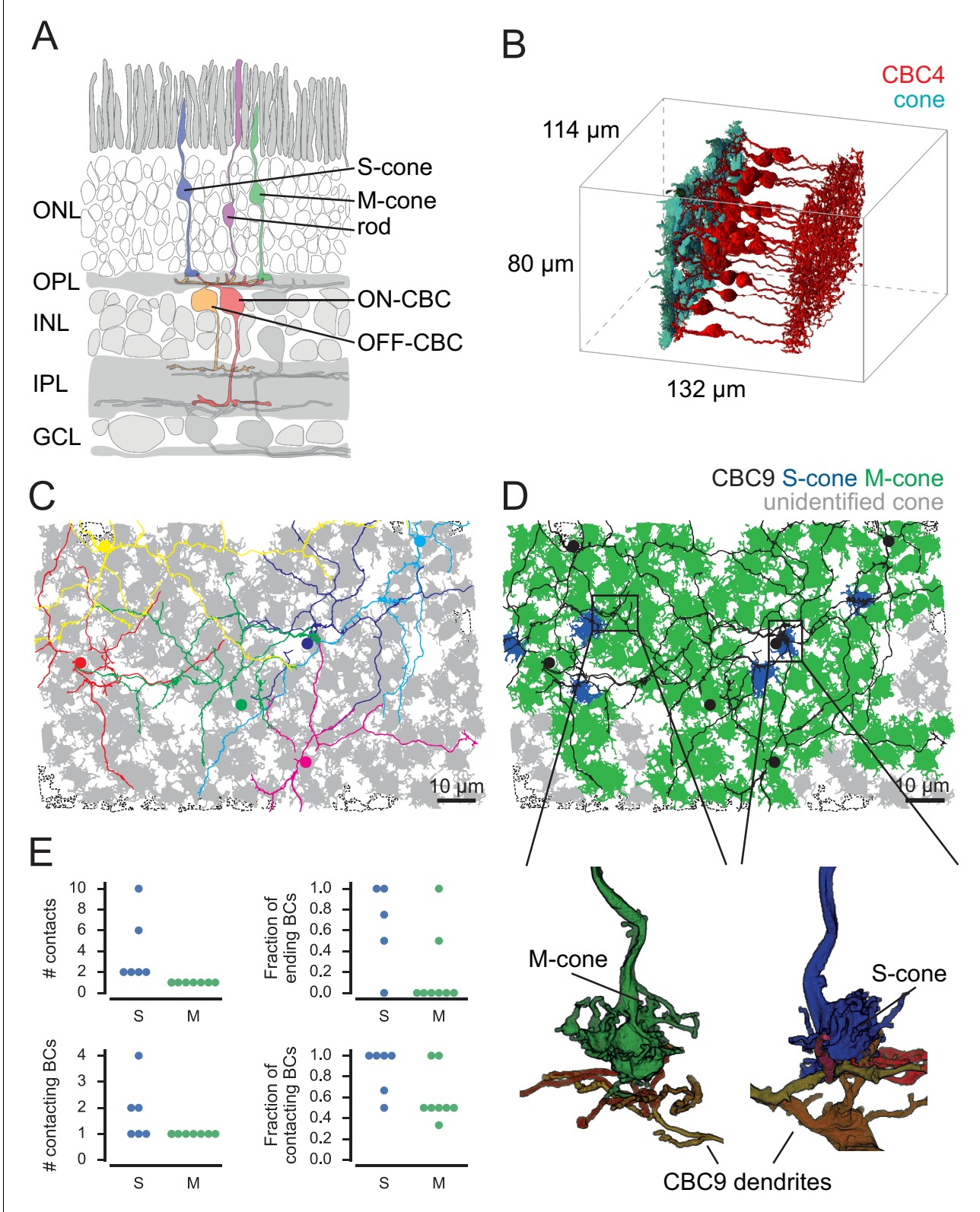

**Figure 1.** Identification of S- and M-cones. (**A**) Scheme showing vertical section through the mouse retina. (**B**) Volume-reconstructed cones and all CBC4 cells. (**C**) Cone pedicles (grey) with CBC9s. BC soma localization is indicated by colored dots. Dashed outlines indicate incomplete cones. (**D**) Same as in **C**, but with putative S-cones (blue) and M-cones (green) highlighted. Unidentified cones are shown in grey. Insets indicate the location of

*Figure 1 continued on next page*

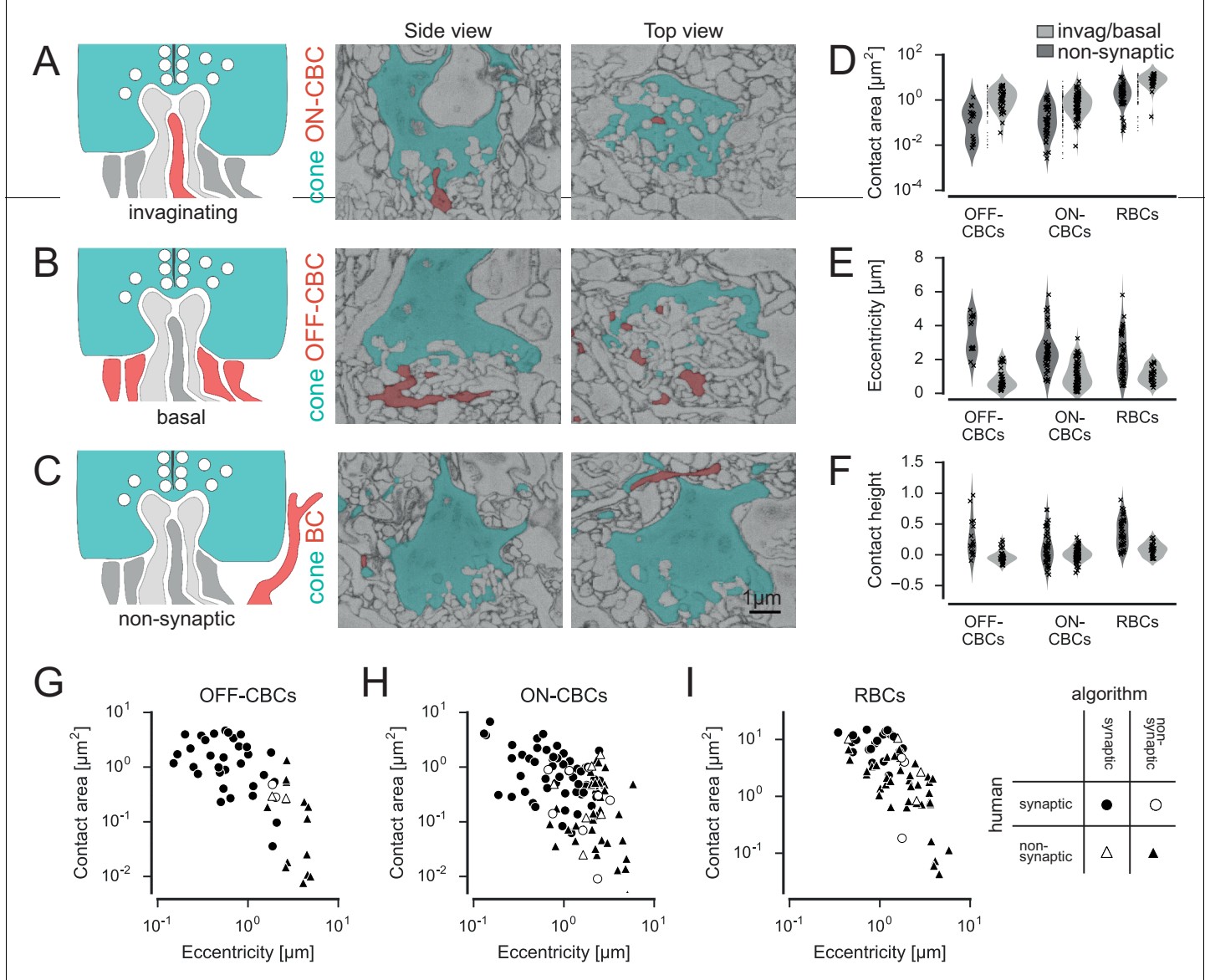

**Figure 2.** Classification of cone contacts. (**A**) Invaginating ON-CBC contact. Schematic drawing (left), EM side view (center) and top view (right). Red and dark grey, BC dendrites; light grey, horizontal cell dendrites; cyan, cone pedicles. (**B**) Basal/flat OFF-CBC contact as in **A**. (**C**) Peripheral (non-synaptic) BC contact as in **A**. (**D–F**) Contact area (**D**) eccentricity (**E**) contact height (**F**) of invaginating/basal and non-synaptic contacts for OFF-/ON-CBCs and rod bipolar cells (RBCs). (**G–I**) Contact area versus eccentricity for OFF-CBC (**G**), ON-CBC (**H**) and RBC (**I**) contacts indicating correctly and incorrectly classified contacts.

The following figure supplements are available for figure 2:

**Figure supplement 1.** Illustration of parameters used for classifying contacts.

**Figure supplement 2.** Examples for disagreements between human and algorithmic classification.

bipolar cells (ON-CBCs) make invaginating contacts, where the dendritic tips reach a few hundred nanometers into the presynaptic area of cone pedicles (*Figure 2A*) (*Dowling and Boycott, 1966*). In contrast, OFF-cone BCs (OFF-CBCs) make basal contacts in the same area (*Figure 2B*). These 'true' contacts have to be distinguished from contacts in the periphery or at the (out)sides

of the cone pedicle as well as contacts between dendrites and cone telodendria, which can happen, for instance as dendrites pass by (*Figure 2C*).

In total, we found n = 20,944 contacts in n = 2620 pairs of cones and BCs. We trained a support vector machine (SVM) classifier to distinguish whether or not an individual BC obtains input from a cone (as opposed to classifying each individual contact site, see Materials and methods). To this end, we defined a set of seven features, including contact area, eccentricity and contact height, which allowed distinguishing between potential synaptic contacts and 'false' contacts (*Figure 2D–F*, *Figure 2—figure supplement 1*). For training of the classifier, we manually labeled a randomly selected set of contacts (n = 50 for OFF-CBCs, n = 108 for ON-CBCs and n = 67 for RBCs). Given the highly stereotypical anatomy of the photoreceptor-BC synapse, labeling performed by an experienced human observer is expected to be very accurate – we here consider therefore the human labels as 'ground-truth'. We trained separate classifiers for ON-CBCs, OFF-CBCs and RBCs and found that they could reliably distinguish between true and false contacts, with a success rate of ~90% (leave-one-out cross-validation accuracy, *Figure 2G–I*). Deviations between the labels of the automatic classifier and the human labels did not vary systematically with BC type (see Materials and methods). Such deviations typically occurred when human labels were assigned based on more global structural features of a contact; such more contextual features were not included in the features used for automatic classification (examples of misclassified contacts are shown in *Figure 2—figure supplement 2* and *Videos 1–3*).

## Contacts between cones and CBCs

We analyzed contacts between CBCs and S- and M-cones in the center of the EM stack where cones were covered by a complete set of all BC types. There was no difference in the number of CBCs contacted by S- and M-cones with 12.2 ± 1.5 CBCs (n = 5 cones, mean ± SEM) for S-cones and 12.2 ± 0.4 CBCs (n = 71 cones) for M-cones, respectively. Similarly, the total number of contact points per cone was almost identical for S- and M-cones with an average of 108 ± 24 per S- and 105 ± 5 per M-cone.

We first studied the convergent connectivity onto the different CBC types and studied how many cones provide input to a single BC of each type (*Figure 3A and B*). To this end, we classified type 5 CBCs, which had not been further subdivided by Helmstaedter et al. (*Helmstaedter et al., 2013*), into three types (*Figure 3—figure supplement 1*, see Materials and methods) in agreement with recent reports (*Greene et al., 2016*).

Most CBC types were contacted predominantly by M-cones, with an average of 2–6 cones contacting individual CBCs. One exception was the CBC9 that – by our definition of S-cones – received considerable S-cone input. We also detected a few contacts between CBC9s and M-cones; these are a consequence of our restrictive definition of S-cone and originate from those cones for which we found only single CBC9 contacts, such that they were classified as M-cone (see above, *Figure 1*; see also *Figure 3—figure supplement 2*).

We next evaluated the divergent connectivity from S- and M-cones to CBCs and studied how many individual BCs of each type were contacted by a single cone (*Figure 3C*). We found that each M-cone contacted on average a little less than one CBC1, while S-cones contacted almost no CBC1, consistent with previous reports (*Breuninger et al., 2011*). Conversely, we found that M-cones almost never contacted CBC9s (see above), but S-cones contacted on average two. Both cone types contacted all other CBC types (*Figure 3D*), with each cone making contact with

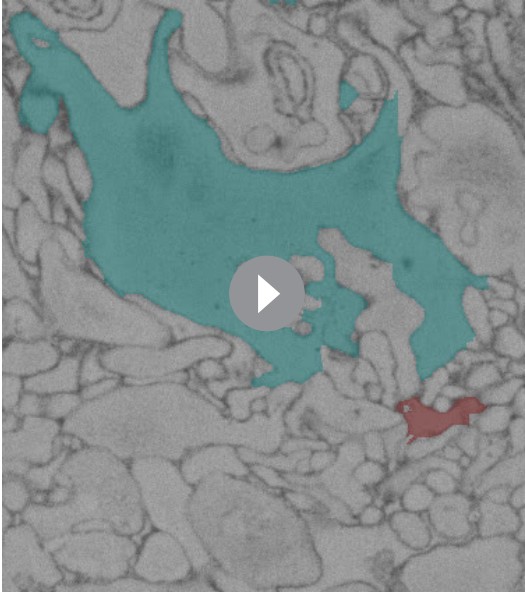

**Video 1.** Animated 3D stack of an ON contact (human: contact, algorithm: no contact).

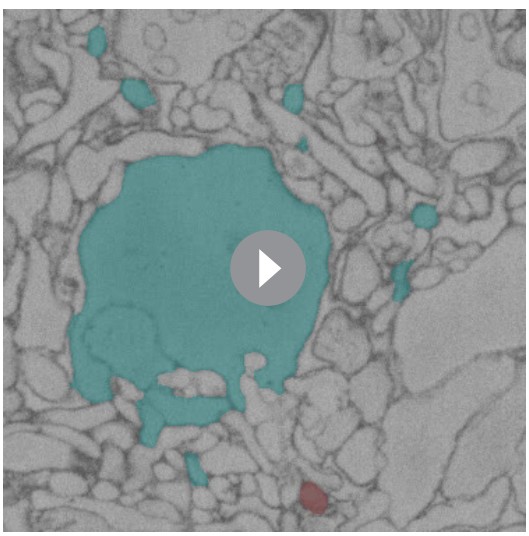

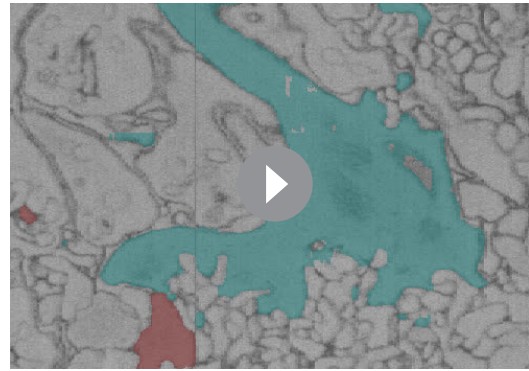

**Video 3.** Animated 3D stack of a RBC contact (human: no contact, algorithm: contact).

**Video 2.** Animated 3D stack of an ON contact (human: no contact, algorithm: contact).

at least one CBC2, 3B, 4, 5I, 6 and 7. In contrast, some cones did not contact ON-CBC types 5T, 5O, X and 8, such that they were contacted by considerably less than one cone on average.

In addition, we tested the hypothesis that CBCs other than type 1 and 9 unselectively contact all cones within their dendritic field (*Wässle et al., 2009*). To this end, we compared the number of contacted cones and the number of cones that are in reach of the BC dendrites (*Figure 3E–G*). OFF-CBCs (types 1–4) contacted on average 65–75% of the cones in their dendritic field, with very similar numbers across types (*Figure 3G*). In contrast, ON-CBCs showed greater diversity: The connectivity pattern of types 5I, 6 and 7 was similar to that observed in the OFF types (*Figure 3G*); these cells sampled from the majority of cones within their dendritic field (60–80%). CBC5T, 5O, X and 8, however, contacted less than half of the cones within their dendritic field (*Figure 3G*), with the lowest fraction contacted by CBCX (~20%). This result is not due to a systematic error in our contact classification: We manually checked volume-reconstructed dendritic trees of the respective types for completeness and frequently found dendrites passing underneath a cone with a distance of 1–3 μm without contacting it (*Figure 3—figure supplement 3*).

Finally, we studied the contact density along CBC dendrites (*Figure 3H and I*). To check for systematic variation independent of the absolute size of the CBC dendritic tree, we normalized the cone contact density for the dendritic field size of each CBC type (*Figure 3I*). Almost all CBC types received input at a very similar location relative to their soma, except for CBCX, which received the majority of inputs closer to the soma than all other types relative to its dendritic field size.

As a control, we also ran the connectivity analysis with the set of S-cones from our alternative, more liberal classification (*Figure 3—figure supplement 2B,C*). In this analysis, CBC9 was the only color specific BC type whereas all other BC types, including CBC1, contacted both S- and M-cones without preferences (*Figure 3—figure supplement 2C*). This contradicts the result of a previous analysis based on physiology, which implies that CBC1 does not receive S-cone input (*Breuninger et al., 2011*).

## The CBCX has few and atypical cone contacts

CBCX had an atypical connectivity pattern compared to other CBC types, so we decided to study its connections in more detail. This BC type has only recently been identified by (*Helmstaedter et al., 2013*; *Shekhar et al., 2016*). It has a compact dendritic tree but a relatively wide axonal terminal system that stratifies narrowly at approximately the same depth as CBC5O and 5I do. Interestingly, CBCX seems to sample the cone input very sparsely, with input from only two cones on average, and contacting only about 20% of the cones available in its dendritic field (*Figure 3C,D and G*). In fact, dendrites of CBCX oftentimes passed underneath cones or even stopped shortly before cone pedicles without making contacts at all (*Figure 4A and B*). It is unlikely that this resulted from

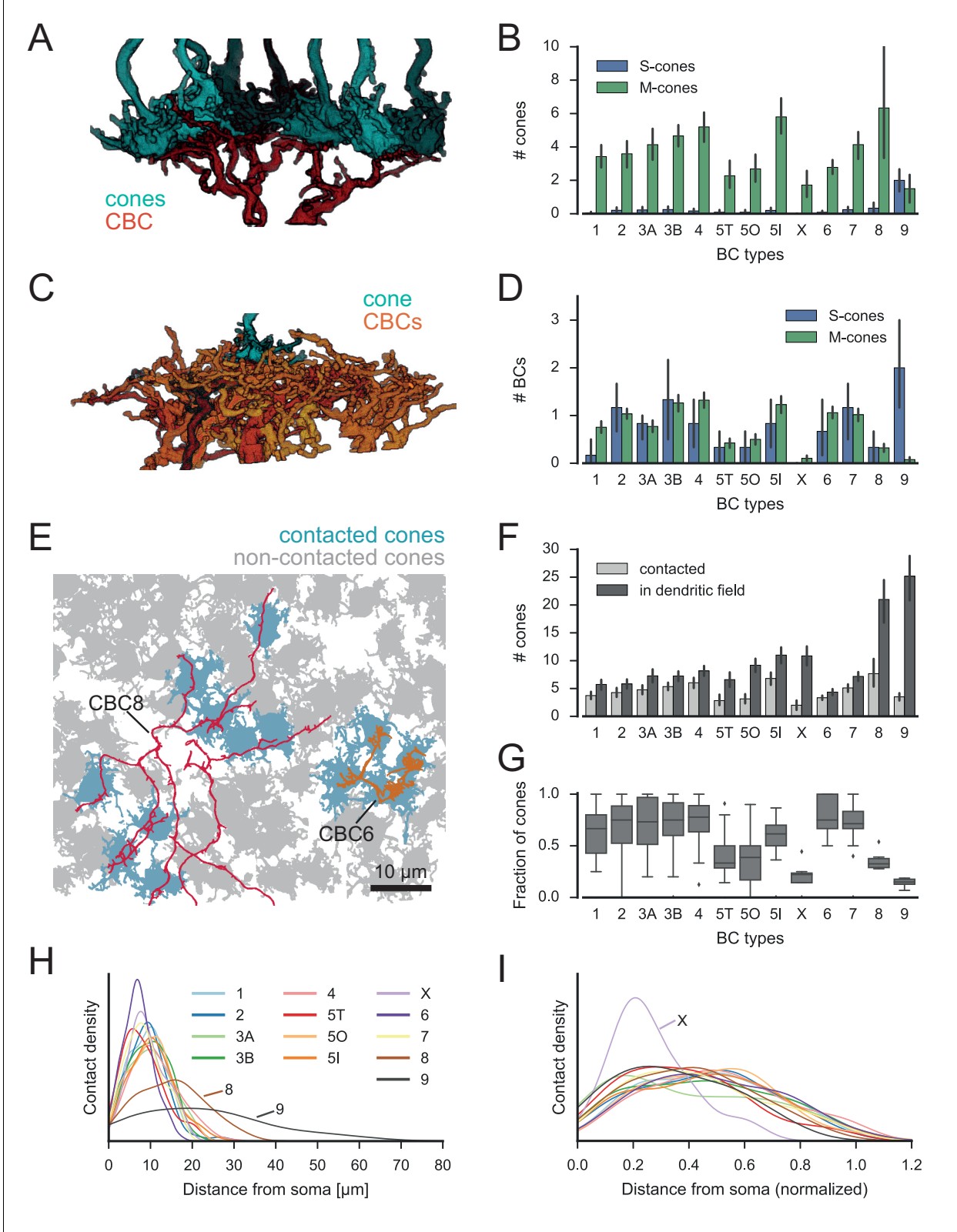

**Figure 3.** Quantification of cone-to-CBC contacts. (A) Volume-reconstructed single BC dendrite (red) contacting numerous cone pedicles (cyan). (B) Number of S- and M-cones contacted by different CBC types. (C) Volume-reconstructed single cone (cyan) contacted by multiple BCs (orange/red). (D) Number of CBCs per type contacted by individual S- and M-cones. (E) Example cone array with CBC6 and CBC8 contacting cones. Grey, non-contacted cones; blue, contacted cones. (F) Number of contacted cones and cones within dendritic field for different CBC types. (G) Fraction of

*Figure 3 continued on next page*

*Figure 3 continued*

contacted cones/cones within the dendritic field. (H) Kernel density estimate of the distribution of contacted cones as function of distance from BC somata. (I) Same as H. but distance normalized by dendritic field size. Bars in **B,D,F** indicate 95% CI.

The following figure supplements are available for figure 3:

**Figure supplement 1.** Classification of type 5 BCs.

**Figure supplement 2.** Connectivity analysis for alternative s-cone classification.

**Figure supplement 3.** Example of a passing dendrite without contacts.

incomplete skeletons for these BCs, as all skeletons were independently verified for this study and corrected where necessary (see Materials and methods).

We re-examined all detected contacts between CBCXs and cones and found that very few of those were 'classical' invaginating ON-CBC contacts (3 out of 19 contacts, n = 7 cells, *Figure 4B–D*). The vast majority were 'tip' contacts (16 out of 19 contacts, n = 7 cells), which were similar to basal contacts made by OFF-CBC dendrites (*Figure 4B–D*). The available data was not conclusive with regards to the question whether these tip contacts of CBCX are smaller than those of OFF-CBCs (median area: 0.05 $\mu m^2$ for n = 22 CBCX contacts; 0.10 $\mu m^2$ for n = 23 OFF-CBC contacts, but p=0.17, Wilcoxon ranksum test).

In contrast to the CBCX, the other ON-CBC types made mostly invaginating contacts (71 out of 81 contacts, n = 12 cells, two cells per BC type, *Figure 4D*), indicating a significant effect of cell type on contact type (GLM with Poisson output distribution, n = 38, interaction: p=$3.6\times10^{-7}$, see Materials and methods). We checked if CBCX receive rod input instead but did not observe any rod contacts (see below). Thus, the CBCX appears to be an ON-CBC with both very sparse and atypical cone contacts similar to those made by OFF-CBCs. Still, based on the axonal stratification depth (*Helmstaedter et al., 2013*) and recent electrophysiological and functional recordings (*Ichinose et al., 2014*; *Franke et al., 2016*) this BC type is most likely an ON-CBC, supported by its mGluR6 expression (*Shekhar et al., 2016*).

## RBCs make contacts with cones

We next analyzed the connectivity between photoreceptors and rod bipolar cells (RBCs) to test the hypothesis that RBCs may contact cones directly (*Pang et al., 2010*). Three cells labeled as RBC in Helmstaedter et al. (*Helmstaedter et al., 2013*) were excluded from this analysis, as they did not contact any rods (*Figure 5—figure supplement 1*). We also found some rods not contacted by any RBC, which is likely due to incomplete tracing of the fine dendritic tips of some RBCs.

In fact, RBCs did not only contact rod spherules but also cone pedicles (*Figure 5A,B*). These contacts were typical ON-CBC contacts with invaginating dendritic tips into the cone pedicles (*Figure 5B*). To quantify the cone-to-RBC connectivity in more detail, we counted the number of cones contacted by an individual RBC. While the vast majority (75%) contacted at least one cone, only 25% of all RBCs (n = 141) did not contact any (*Figure 5C*). However, we did not find a preference of RBCs to connect S- or M-cones (*Figure 5D*). Conversely, 45% of cones contacted a single RBC, ~35% spread their signal to two to four RBCs, and only 20% of the cones did not make any contact with an RBC (*Figure 5E*). Our finding provides an anatomical basis to the physiologically postulated direct cone input into a subset of RBCs (*Pang et al., 2010*). Next, we evaluated whether RBCs contacting only rods or both cone(s) and rods represent two types of RBC, as hypothesized by *Pang et al. (2010)*. However, the two groups of RBCs did not differ regarding the stratification depth of their axonal arbor (*Figure 5—figure supplement 2A*), number of rod contacts (*Figure 5—figure supplement 2B*) or potential connectivity to AII amacrine cells (*Figure 5—figure supplement 2C*), and did not form independent mosaics (*Figure 5—figure supplement 2D*). In addition, the dendritic field size (116 vs 131 $\mu m^2$, p=0.1, n = 139 RBCs) and the number of dendritic tips (46 vs. 45.5, p=0.8, n = 12 manually counted RBCs) did not differ significantly between the two groups. Therefore, the available anatomical data argue against two types of RBC.

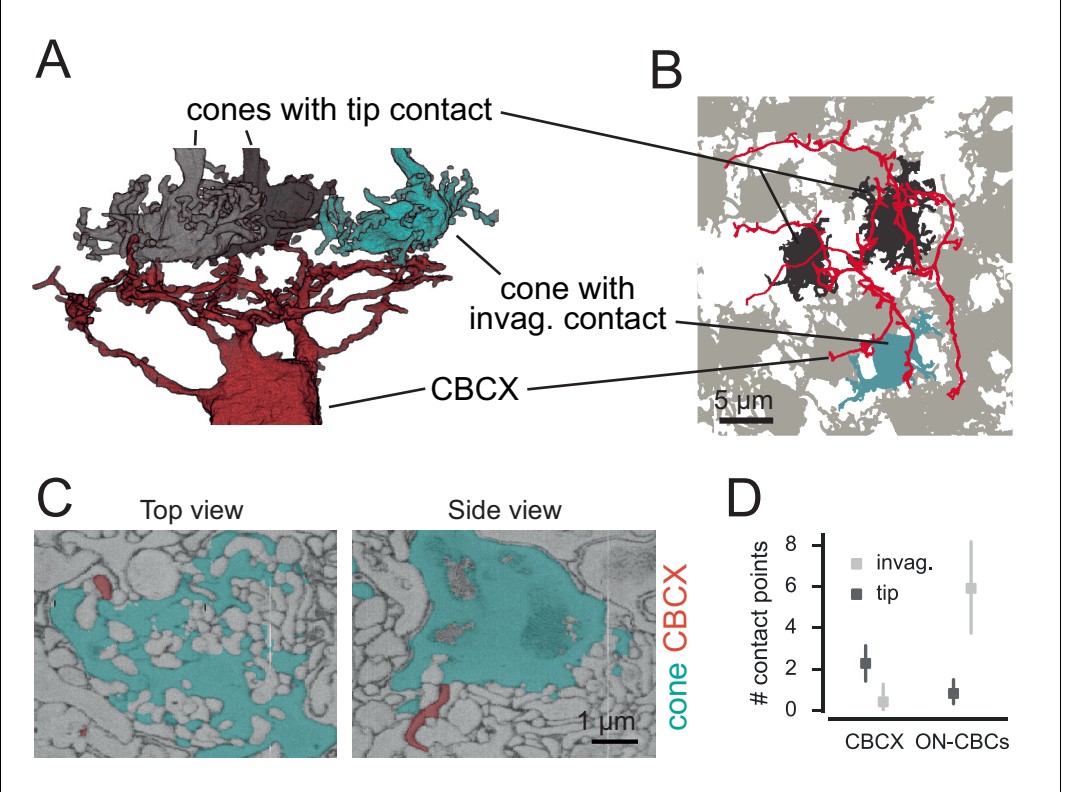

**Figure 4.** CBCX makes few and atypical cone contacts. (**A**) Volume-reconstructed CBCX dendritic arbor (red) contacting few cone pedicles (cyan, invaginating contact; grey, tip contact). (**B**) Same exemplary cone array as in **A**. with CBCX dendritic arbor contacting cones. Light grey, non-contacted cones; cyan, invaginating contacts; dark grey, tip contacts. (**C**) EM image showing tip contact between CBCX (red) and cone pedicles (cyan), top view (left) and side view (right). (**D**) Invaginating and tip contacts in CBCXs and other ON-CBCs. Bars in **D**. indicate 95% CI.

## Quantification of rod to OFF-CBC contacts

Analogous to the analysis above, we skeletonized and volume rendered a complete set of over 2000 neighboring rod spherules (about 50% of the EM field, *Figure 6A*, *Figure 6—figure supplement 1*) and identified rod-to-bipolar cell connections. In addition to the well-described invaginating rod-to-RBC connections (*Figure 6B*), we also found basal contacts between OFF-CBCs and rods close to the invaginating RBC dendrites (*Figure 6C*), as described earlier (*Hack et al., 1999*; *Mataruga et al., 2007*; *Haverkamp et al., 2008*; *Tsukamoto and Omi, 2014*). We did not find any contacts between ON-CBCs and rods (in agreement with *Tsukamoto and Omi, 2014*; but see *Tsukamoto et al., 2007*).

A single RBC contacted about 35 rods (*Figure 6D*), which is slightly more than what was recently reported (~25 rods, *Tsukamoto and Omi, 2013*). A single rod contacted one or two RBCs, but very rarely no RBC or more than two (*Figure 6E*). In all cases with two invaginating dendrites, the dendrites belonged to two different RBCs (n = 30 rods). The rods without RBC contacts were mainly located at the border of the reconstructed volume, where we could not recover all RBCs. The number of rods contacting OFF-CBCs was much lower: Whereas CBC1 and CBC2 did not receive considerable rod input, CBC3A, CBC3B and CBC4 were contacted by 5–10 rods, with CBC3B receiving the strongest rod input (*Figure 6D*).

## Discussion

We analyzed an existing electron microscopy dataset (*Helmstaedter et al., 2013*) to quantify the connectivity between photoreceptors and bipolar cells. We found interesting violations of

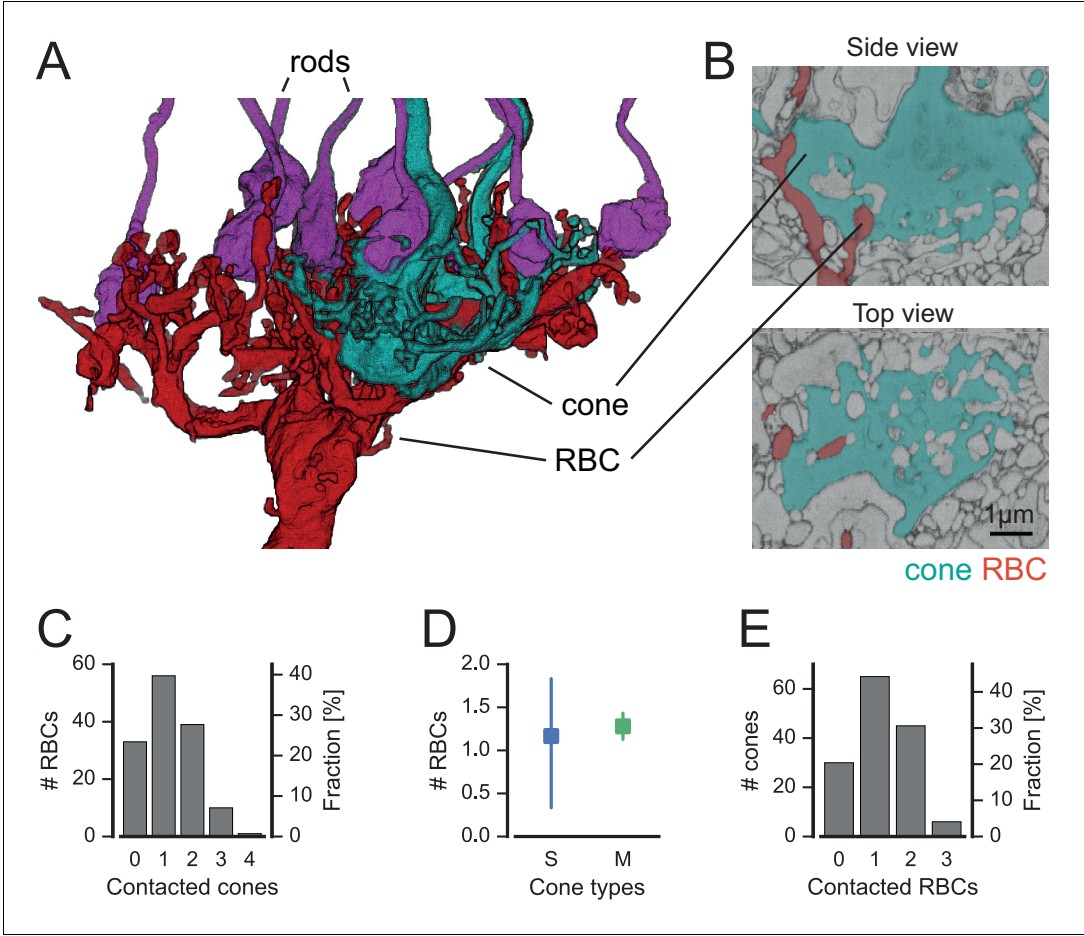

**Figure 5.** Cones contact rod bipolar cells. (**A**) Volume-reconstructed RBC (red) contacting both rods (magenta) and cone pedicles (cyan). (**B**) EM images showing invaginating contact between cone (cyan) and RBC (red), side view (top) and top view (bottom). (**C**) Number of RBCs contacted by cones. (**D**). Number of RBCs contacted by S- and M cones. (**E**) Number of cones contacted by RBCs. Bars in **D**. indicate 95% CI.

The following figure supplements are available for figure 5:

**Figure supplement 1.** Excluded RBCs.

**Figure supplement 2.** No evidence for two RBC subtypes.

established principles of outer retinal connectivity: The newly discovered CBCX (*Helmstaedter et al., 2013*), likely an ON-CBC (*Ichinose et al., 2014*; *Franke et al., 2016*), had unexpectedly few and mostly atypical basal contacts to cones. While CBC types 5T, 5O and 8 also contacted fewer cones than expected from their dendritic field, they exhibited 'standard' invaginating synapses. In addition, we provide anatomical evidence that rod and cone pathways are interconnected, showing frequent cone-RBC contacts. The emerging picture of BC types with their input profiles are summarized in *Figure 7*.

## Does a 'contact' represent a synaptic connection?

Since the dataset we used was not labeled for synaptic structures, we used automatic classifiers based on structural criteria to identify putative synaptic contacts between BCs and photoreceptors. Due to the highly stereotypical anatomy of the photoreceptor-BC synapse, these criteria allow unambiguous identification of synaptic sites for trained humans (see also *Results*). For example, we used the proximity of the closest contact to the center of the cone pedicle region as a feature, where

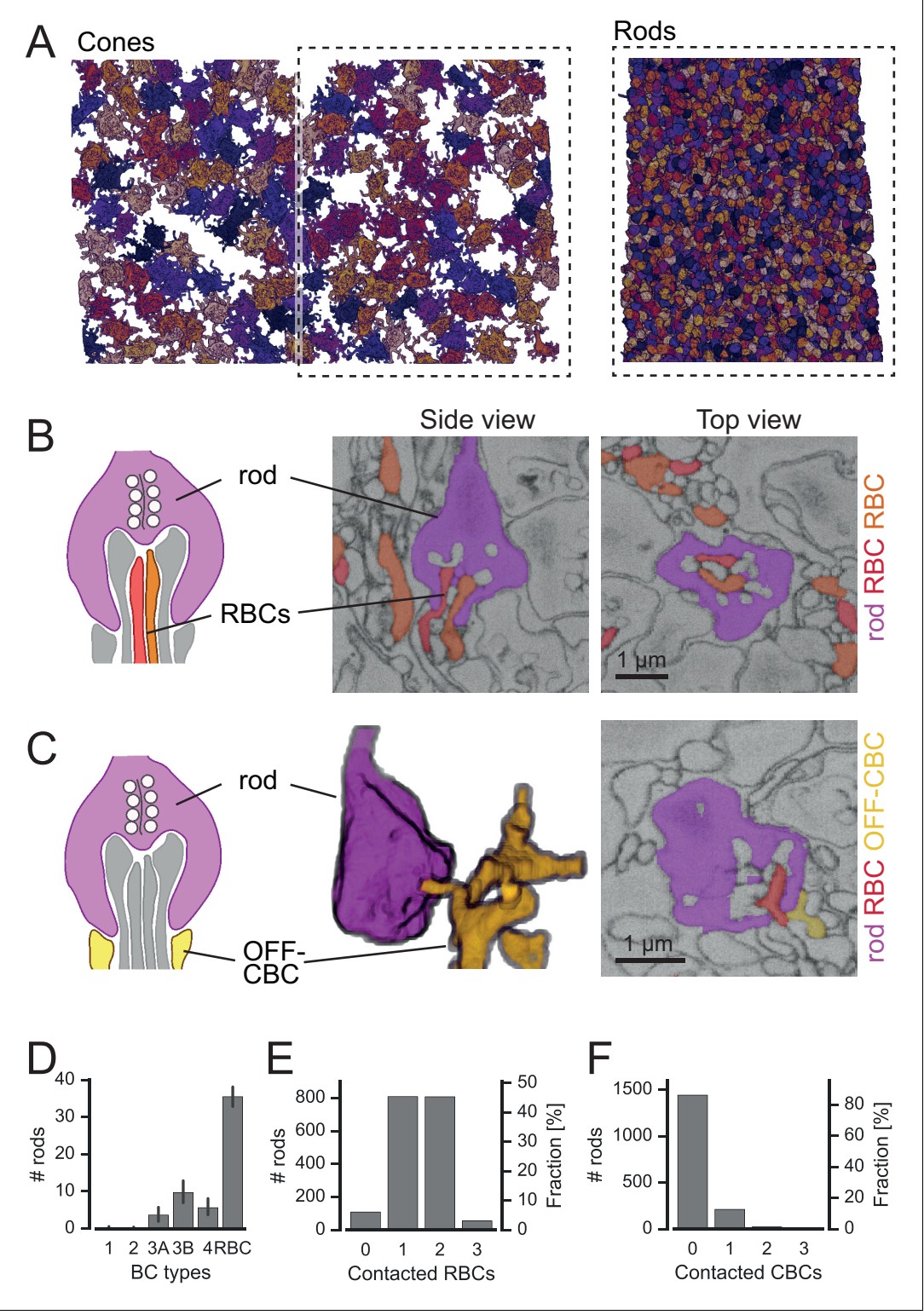

**Figure 6.** Rods contact RBCs and OFF-CBCs. (**A**) Volume-reconstructed, neighboring rod spherules (right) in one half of the field of the reconstructed cone pedicles (left). (**B**) Rod spherule (magenta) with invaginating dendrites of two RBCs (orange, red). Schematic drawing (left), EM images side view (middle) and top view (right). (**C**) Rod spherule (magenta) with basal contacts by OFF-CBCs (yellow). Schematic (left), volume-reconstructed vertical view (middle), EM image with top view (right). The latter also shows an invaginating RBC dendrite (red). (**D–F**). Number of rods (and fraction) contacted by RBCs (**D,E**), and OFF-CBC types (**D, F**). Bars in **D**. indicate 95% CI.

*Figure 6 continued on next page*

*Figure 6 continued*

The following figure supplement is available for figure 6:

**Figure supplement 1.** Classification of rod contact classification.

presynaptic ribbons have been reported at the ultrastructural level (*Dowling and Boycott, 1966*; *Chun et al., 1996*).

The overall accuracy of the classifiers evaluated with human annotated labels was high (~90%). Nevertheless, it is possible that a few contacts were misclassified. Manual quality control, however, revealed no systematic errors. Therefore it is unlikely that classification errors affected our main conclusions. Mismatches between human and classifier labels usually occurred when the human used the global context for the assessment of a contact, knowledge that is not easily transferred into an algorithm (*Figure 2—figure supplement 2*). Still, human error cannot be ruled out: For instance, contact points labeled by humans as non-synaptic may feature a gap junction and are therefore functional rather than random. For reference, all data including software for classifying and examining BC-cone contacts is available online.

## Is there an effect of retinal location?

Unfortunately, the retinal location of the EM stack used here is unknown (*Helmstaedter et al., 2013*); it may originate from the ventral retina, where M-cones co-express S-opsin (*Röhlich et al., 1994*; *Baden et al., 2013*) However, as 'true' S-cones were shown to be evenly distributed across the retina (*Haverkamp et al., 2005*), CBC9 connectivity can be used for identification of S-cones independent of location. Nevertheless, it cannot be excluded that opsin co-expression in M-cones in the ventral retina may influence the connectivity patterns between the M-cones and the remaining bipolar cell types.

## Sparse contacts between some ON CBC types and cones

We found that ON-CBCs 5T, 5O, X and 8 contact fewer cones than expected from the size of their dendritic field. We observed that many of their dendrites passed by the cone pedicles with a distance of 1–3 μm or even ended under a cone pedicle without contacting it (*Figure 3—figure*

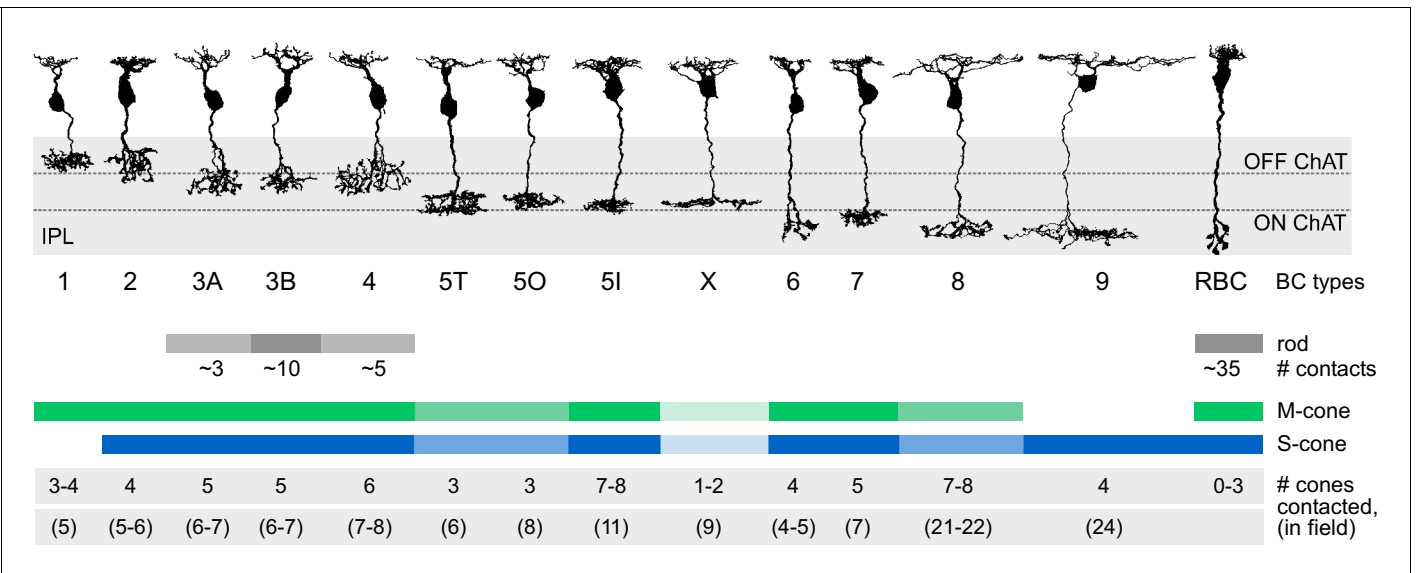

**Figure 7.** Connectivity between cone and rod photoreceptors and bipolar cells in the mouse retina. Representative examples of bipolar cell types in the mouse retina are shown. The number of cones in the dendritic field number and contacted photoreceptors are given for each type.

*supplement 1*). This is in agreement with a recent study reporting that CBC8 does not contact all cones within its dendritic field (*Dunn and Wong, 2012*), but in contrast to earlier studies that concluded that different diffuse BC types receive input from all cones within their dendritic field (*Boycott and Wässle, 1991*; *Wässle et al., 2009*). However, a crucial difference with the earlier studies and our study is spatial resolution: Conventional light microscopy can resolve depth with a resolution of several hundreds of nanometers, whereas the EM dataset we used has a resolution of 25 nm, allowing us to more accurately assess whether pre- and postsynaptic structures are in contact with each other.

Could diffusion-based synaptic signaling play a role in ON-CBCs with sparse contacts? 'Diffusion-based' synaptic contacts have been proposed for OFF-CBCs (*DeVries et al., 2006*) and between cones (*Szmajda and Devries, 2011*). However, although diffusion-based transmission may be present in the outer retina, there is no evidence so far that cone signals reach bipolar cells that neither make invaginating or basal contacts (i.e. with dendrites just passing by a cone pedicle).

## CBCX makes atypical contacts with cones

As shown above, the CBCX makes the fewest contacts with cones of all ON CBCs. On average, they contacted only about two cones, representing only 20% of the cones within the area of their dendrites. This finding is in agreement with a recent single-cell RNA-seq study, which found that CBCXs feature lower expression levels for metabotropic glutamate receptor mGluR6 (*grm6*) – the hallmark of ON-BCs – compared to other ON-CBC types (*Shekhar et al., 2016*). The mouse CBCX is reminiscent of the giant CBC in macaque retina (*Joo et al., 2011*, *Tsukamoto and Omi, 2016*) with respect to several features: Also the latter has a very large and sparsely branched dendritic tree and a relatively large axonal arbor that stratifies in the middle of the IPL and contacts only about 50% of the cones in its dendritic field.

In contrast to all other ON-CBCs, we found that the vast majority of CBCX contacts were not invaginating but rather resembled basal OFF-CBC contacts. It is unclear if these tip contacts are indeed functional synaptic sites. This is not the first finding to challenge the traditional view that ON-CBCs form only invaginating and OFF-CBCs only basal synaptic contacts. In the primate fovea, diffuse ON-CBCs (DBs) form basal contacts with foveal cones since almost all invaginating sites are taken by midget bipolar cell dendrites (*Calkins et al., 1996*). This spatial limitation is less evident in mid-peripheral primate retina. At 3–4 mm eccentricity, diffuse ON-CBCs receive 10% (DB5) to 40% (DB4 and DB6) of their cone input through basal synapses (*Hopkins and Boycott, 1996*).

Interestingly, CBCX contacts in the IPL also appear to be distinct from those of other BC types: First, the majority of cells contacted by CBCX in the IPL are amacrine cells rather than ganglion cells (*Helmstaedter et al., 2013*). Second, they form sparse contacts relative to their axon terminal size with comparatively few cells. Thus, the CBCX seems to be an exception, an unusual BC type in many respects in addition to its sparse and atypical connectivity properties in the OPL, reminiscent of a recently described dendrite-less interneuron type that expresses BC-specific genes (*Shekhar et al., 2016*) and was named GluMI (glutamatergic monopolar interneuron) (*Della Santina et al., 2016*). It is conceivable that – similar to the GluMI – the CBCX is evolutionary on its way to retracting its dendrites. Alternatively, CBCX develops a tad later than other CBC types and most potential synaptic sites at cone ribbons are already occupied, such that the CBCX can only form few connections – reminiscent of the situation in primate (see above).

## May RBCs form an additional photopic ON channel?

We found that cones connect to 75% of RBCs; in many cases, one cone contacted multiple RBCs. In turn, 35% of RBCs received converging input from several cones. This massive cone input via invaginating synapses to RBCs suggests a prominent use of the primary rod pathway (*Bloomfield and Dacheux, 2001*) during photopic conditions. Consistent with our findings, RBCs can be activated under photopic light conditions (*Franke et al., 2016*). However, since rods recover at high light levels (*Blakemore and Rushton, 1965*), the functional significance of cone input to RBC remains unclear. In principle, it is possible that the observed cone-RBC synapses are developmental 'leftovers' without physiological relevance, but as the cone-RBC contacts look like standard cone-CBC invaginating synapses, we think that it is more likely that they contribute to RBC activation under

photopic conditions, especially in mid-range light intensities where cones are active but rods not yet recovering (*Tikidji-Hamburyan et al., 2015*).

If this was the case, cone activation of RBCs could indirectly inhibit OFF-CBCs via AII amacrine cells. This suggests that RBCs may contribute to crossover inhibition (*Molnar and Werblin, 2007*). On the other hand, it is unclear whether gap junctions between AIIs and ON-CBCs are in an open or closed state under light-adapted conditions (*Bloomfield et al., 1997*; *Kuo et al., 2016*). With open gap junctions, activating RBCs may boost the signal in ON-CBC axon terminals and therefore enhance contrast (in complement with the OFF-CBC inhibition).

Based on the physiological finding that only a subset of RBCs receive input from cones, *Pang et al. (2010)* suggested that there may be two distinct RBC types, with the rod-only one having axon terminals ending closer to the ganglion cell layer. Our data do not provide evidence for two RBC types based on the connectivity in the outer retina. This agrees well with recent findings from single-cell RNA-seq experiments, where all RBCs fell into a single genetic cluster with little heterogeneity (*Shekhar et al., 2016*).

### OFF CBC types contact different numbers of rods

We quantified the number of rods contacting the five OFF-CBC types. Whereas CBC1 and 2 received almost no rod input, we observed flat/basal contacts between rods and types CBC3A, 3B and 4, providing a quantitative confirmation of this finding (*Mataruga et al., 2007*; *Haverkamp et al., 2008*; *Tsukamoto and Omi, 2014*). CBC3A and four received input from ~5 rods in addition to the ~5 cones contacted by them. CBC3B sampled from the same number of cones but was contacted by about twice as many rods. Since these basal contacts between rods and OFF CBCs have been shown to express AMPA receptors (*Hack et al., 1999*), rods likely provide considerable input to OFF-CBCs, possibly representing a distinct scotopic OFF channel complementing the scotopic ON channel via RBCs. Interestingly, the morphologically similar CBC3A and 3B may obtain their (functional) differences not only from the expression of different ionotropic glutamate receptors (*Puller et al., 2013*) but also from their connectivity with rods.

### Conclusion

Here, we performed a systematic quantitative analysis of the photoreceptor-to-bipolar cell synapse. We showed that there are exceptions to several established principles of outer retinal connectivity. In particular, we found several ON-BC types that contacted only a relatively small fraction of the cones in their dendritic field. We also find that rod and cone pathways already interact strongly in the outer plexiform layer. Whether these are general features of mammalian retinas or evolutionary specializations unique to the mouse remains to be seen.

## Materials and methods

### Dataset and preprocessing

We used the SBEM dataset e2006 published by (*Helmstaedter et al., 2013*) for our analysis (http://www.neuro.mpg.de/connectomics). The dataset has a voxel resolution of 16.5×16.5×25 nm with dimensions 114 µm × 80 µm × 132 µm. We performed volume segmentation of the outer plexiform layer (OPL) using the algorithms of (*Helmstaedter et al., 2013*). The preprocessing of the data consisted of three steps: (i) Segmentation of the image stack, (ii) merging of the segmented regions and (iii) collection of regions into cell volumes based on traced skeletons.

We modified the segmentation algorithm to prevent merging of two segments if the total volume was above a threshold (>50,000 voxels), as sometimes the volumes of two cone pedicles could not be separated with the original algorithm. Although this modification resulted in overall smaller segments, these were collected and correctly assigned to cells based on the skeletons in the last step of the preprocessing.

We identified 163 cone pedicles and created skeletons spanning their volume using the software KNOSSOS ([*Helmstaedter et al., 2012*], www.knossostool.org,). We typically traced the center of the cone pedicle coarsely and added the individual telodendria for detailed reconstruction. In addition, we traced 2177 rod spherules covering half of the dataset (*Figure 6*). For our analysis, we used all photoreceptors for which at least 50% of the volume had been reconstructed (resulting in 147

cones and 1799 rods). We used the BC skeletons published by *Helmstaedter et al. (2013)*, with the following exceptions: We completed the dendritic trees of three XBCs (CBCXs), which were incompletely traced in the original dataset. In addition, we discarded three BCs originally classified as RBCs because they were lacking rod contacts as well as the large axonal boutons typical for RBCs (Supp. *Figure 6A–C*), and one BC classified as a CBC9 because its dendritic field was mostly outside of the data stack (Supp. *Figure 6D*).

Next, we used the algorithm by (*Helmstaedter et al., 2013*) to detect and calculate the position and area of 20,944 contact points between cone pedicles and BC dendrites and 7993 contact points between rod spherules and BC dendrites. To simplify the later visual inspection of contacts, we used the reconstructed cell volumes to generate colored overlays for the raw data to highlight the different cells in KNOSSOS.

## Identification of S-cones

We detected 169 contacts in 51 pairs of CBC9s and cones. Upon manual inspection, we found a total of 32 invaginating (potentially synaptic) contacts between 6 CBC9s and 14 cone pedicles.

Based on immunocytochemistry, it has been shown that S-cones are contacted by all CBC9 within reach and that CBC9 contacts to S-cones are mostly at the tips of the dendrites (*Haverkamp et al., 2005*). For all 14 contacted cones, we analyzed the number of invaginating CBC9 contacts, the number of contacting CBC9s, the fraction of CBC9 with dendrites close to the cone that make contact and whether the dendrites end at the cone or continue beyond it (*Figure 1E*). Based on these criteria, we classified 6 out of these 14 cones as S-cones (see also *Figure 1—figure supplement 1*). In addition to our main analysis, we present an alternative analysis that considers the case if all 14 cones were counted as S-cones (*Figure 3—figure supplement 2*).

## CBC5 classification

CBC5s were classified initially based on their connectivity to ganglion cells and amacrine cells into types 5A and 5R, where 5R was a group containing multiple types (*Helmstaedter et al., 2013*). In addition, some CBC5s could not be classified due to a lack of axonal overlap with the reconstructed ganglion cells of the types used for classification. Considering the separate coverage factors for dendritic and axonal overlap of all CBC5s together (OPL: 3.14, IPL: 2.89), dividing them into three subtypes is conceivable considering the numbers for other CBC types (*Table 1*). This has already been suggested by (*Greene et al., 2016*), who divide CBC5s into three subtypes based on axonal density profiles (using a different EM dataset that includes only the inner retina).

We followed the classification approach suggested by Greene et al. (*Greene et al., 2016*): First, we calculated the densities of both ON- and OFF-starburst amacrine cells (SACs) dendrites along the optical axis. We fitted the peak of these profiles with a surface using bivariate B-splines of third order. Next, we corrected the density profiles of CBC5 axonal trees by mapping the SAC surfaces to parallel planes. We then applied principal component analysis (*Figure 3—figure supplement 1A*) to obtain a first clustering into three groups by fitting a Gaussian mixture model (GMM) (*Bishop, 2006*) with three components onto the first three principal components of the axon density profiles. The resulting density profiles of the three clusters matches those found by (*Greene et al., 2016*) (*Figure 3—figure supplement 1B*). As we noted a few violations of the postulated tiling of the retina by each type (*Seung and Sümbül, 2014*), we implemented a heuristic to shift cells to a different cluster or swap pairs of cells optimizing a cost function including both overlap in IPL and OPL as well as the GMM clustering (*Figure 3—figure supplement 1*):

$$\mathcal{L} = \lambda_1 \sum_i \sqrt{(x_i - \mu_{c_i})^T \Sigma_{c_i} (x_i - \mu_{c_i})} + \lambda_2 \frac{\sum_{i,j} \delta_{c_i,c_j} O_{ij,OPL}}{\sum_i A_{i,OPL}} + \lambda_2 \frac{\sum_{i,j} \delta_{c_i,c_j} O_{ij,IPL}}{\sum_i A_{i,IPL}}$$

with $x_i$ the parameter vector of cell $i$, $c_i$ the mixture component cell $i$ is assigned to, $\mu_c$ the mean of the mixture component $c$, $\Sigma_c$ the covariance matrix of the mixture component $c$, $\delta_{ij}$ the Kronecker delta, $A_{i,OPL/IPL}$ the area of the dendritic field/axonal tree of cell $i$ and $O_{ij,OPL/IPL}$ the overlap of cell $i$ and $j$ in the OPL/IPL. The overlap of two cells is calculated as the intersection of the convex hull of the dendritic fields/axonal trees. Likely, our CBCX corresponds to CBC5D from (*Shekhar et al., 2016*) and CBC5T to CBC5C. Possibly, CBC5I corresponds to CBC5A and CBC5O to CBC5B (see discussion in *Greene et al., 2016*; *Shekhar et al., 2016*).

**Table 1.** OPL hull area: Average area of convex hull of dendritic field in OPL per cell type [μm$^2$], mean ± SEM; OPL cov.: coverage factor derived from convex hulls by computing the sum of convex hull areas divided by area of the union of convex hulls; OPL cov. cones: coverage factor computed from cones by computing the sum of the number of cones in the dendritic field of each cell divided by the number of cones in the joint dendritic field; Wässle: coverage values from **Wässle et al. (2009)** computed by the same method as OPL cov. cones; IPL hull area: Average area of convex hull of the axonal field in IPL per cell type [μm$^2$], mean ± SD; IPL cov: analogous to OPL cov.

| Type | N | OPL hull area [μm$^2$] | OPL cov. | OPL cov. cones | Wässle | IPL hull area | IPL cov. |
|------|---|------------------------|----------|----------------|--------|---------------|----------|
| CBC1 | 26 | 175 ± 16 | 1.17 | 1.48 | 1.48 | 376 ± 16 | 1.52 |
| CBC2 | 34 | 204 ± 19 | 1.18 | 1.55 | 1.5 | 353 ± 23 | 1.52 |
| CBC3A | 22 | 273 ± 28 | 1.17 | 1.37 | 1.25 | 308 ± 28 | 1.21 |
| CBC3B | 32 | 292 ± 19 | 1.41 | 1.90 | 1.55 | 224 ± 9 | 1.24 |
| CBC4 | 30 | 302 ± 20 | 1.32 | 1.86 | 1.6 | 274 ± 23 | 1.33 |
| CBC5T | 22 | 256 ± 30 | 1.13 | 1.30 | - | 402 ± 25 | 1.28 |
| CBC5O | 22 | 380 ± 41 | 1.35 | 1.60 | - | 359 ± 23 | 1.17 |
| CBC5I | 25 | 459 ± 30 | 1.55 | 1.95 | - | 276 ± 14 | 1.22 |
| CBCX | 7 | 433 ± 34 | 1.02 | 1.12 | - | 899 ± 126 | 1.12 |
| CBC6 | 45 | 125 ± 11 | 1.14 | 1.58 | - | 165 ± 11 | 1.17 |
| CBC7 | 29 | 254 ± 18 | 1.22 | 1.65 | 1.3 | 274 ± 11 | 1.16 |
| CBC8 | 6 | 1249 ± 144 | 1.14 | 1.21 | - | 699 ± 55 | 1.02 |
| CBC9 | 6 | 2223 ± 227 | 1.84 | 1.45 | - | 1605 ± 335 | 1.43 |
| RBC | 141 | 128 ± 3 | 2.17 | 4.37 | - | 65 ± 3 | 1.40 |

## Automatic contact classification

To distinguish potential synaptic contacts between photoreceptors and BCs from accidental contacts, we developed an automatic classification procedure exploiting the stereotypical anatomy of cone-BC synapses (triads, *Dowling and Boycott, 1966*). First, we grouped all contacts for a specific cone-BC pair, in the following referred to as a contact-set. We obtained a training data set by randomly selecting 10 contact-sets per CBC type and 50 RBC-cone contact-sets. We excluded CBCX from the training data because of their atypical contacts. To increase classifier performance we added 17 additional RBC-cone contact-sets manually classified as invaginating contacts as well as all 48 CBC9-cone contact-sets classified for the S-cone identification. For those contact-sets, we visually inspected each individual contact point in the raw data combined with volume segmentation overlay using KNOSSOS. Then we classified it either as a central basal contact (potentially synaptic) or peripheral contact (e.g. at the side of a cone or contact with telodendria, likely non-synaptic) for OFF-CBCs or as invaginating contact vs. peripheral contact for ON-CBCs and RBCs. Next, we extracted a set of seven parameters for each contact (see *Figure 2—figure supplement 1*):

- <u>Contact area</u>: The total contact area aggregated over all contact points between a BC and a cone
- <u>Eccentricity</u>: The distance between the cone center and the closest contact point in the plane perpendicular to the optical axis
- <u>Contact height</u>: The distance of the contact point with minimal eccentricity from the bottom of the cone pedicle (measured along the optical axis, normalized by the height of the cone pedicle).
- <u>Distance to branch point</u>: Minimal distance between a contact point and the closest branch point, measured along the dendrite
- <u>Distance to tip</u>: Minimal distance between a contact point and the closest dendritic tip. A large distance occurs for example for a contact between a passing dendrite and a cone.
- <u>Smallest angle</u> between the dendrite and the optical axis at a contact point
- <u>Number of contact points</u> between cone and BC

**Table 2.** Cross validation results of BC-to-cone contact classification.

| | False positive | False negative | Total score |
|---|---|---|---|
| OFF-CBCs | 12.5% | 5.9% | 0.92 |
| ON-CBCs | 14.0% | 12.3% | 0.87 |
| RBCs | 9.3% | 12.5% | 0.90 |

Based on those parameters we trained a support vector machine classifier with radial basis functions (C-SVM) for each OFF-CBC, ON-CBC and RBC cone contact using the Python package *scikit-learn*. Optimal parameters were determined using leave-one-out cross validation (see *Table 2* for scores and error rates). Typically, 0–2 errors for 10 labeled training samples occurred per BC type (three in one case, CBC3A; 7/48 for CBC9).

### Analysis of rod contacts

As the reconstructed rod spherules cover only half of the EM dataset, we restricted the analysis to bipolar cells with their soma position inside this area. To automatically classify the contacts to rods, we followed a similar scheme as for the cones. Again, we grouped the contacts for each pair of BC and rod spherule. As training data, we selected all putative contact sites with CBC1s (n = 5) and CBC2s (n = 32), 20 random contacts to CBC types 3A, 3B and four as well as 100 random contacts to RBCs. Again, we classified these contacts by visual inspection in KNOSSOS using the raw data with a colored segmentation overlay. In addition, we manually inspected all 132 contact points between rod spherules and ON-CBCs, but could not identify a single potential synaptic contact. We trained SVM classifiers for contacts between rods and RBCs/OFF-CBCs using the same parameters as for the contacts to cones. As synaptic contacts between OFF-CBCs and rod spherules are basal contacts situated close to the invaginating RBC contacts, we added the minimum distance to the next (synaptic) RBC contact as an additional classification parameter for OFF-CBCs. As a consequence, we restricted the analysis of OFF-CBC-to-rod contacts to those rods were RBC contacts could be identified (n = 1685). See *Table 3* for scores and error rates from the leave-one-out cross-validation.

### Statistics

Error bars in all plots are 95% confidence intervals (CI) calculated as percentiles of the bootstrap distribution obtained via case resampling. In *Figure 4D*, we used a generalized linear mixed model with Poisson output distribution and fixed effects contact type and cell type and random effect cell identity (R package *lme4*). The model yielded a significant intercept (z = 8.72, p<0.0001), a significant main effect of cell type (z = 4.11, p=$4x10^{-4}$), a significant main effect of contact type (z = 2.66, p=0.008) and a significant interaction cell x contact type (z = $-5.09$, p<$3.6x10^{-7}$).

### Data and code availability

Jupyter notebooks and data for reproducing our analysis and main figures are available online at https://github.com/berenslab/pr_bc_connectivity.

**Table 3.** Cross validation results of BC-to-rod contact classification.

| | False positive | False negative | Total score |
|---|---|---|---|
| OFF-CBCs | 18.3% | 22.5% | 0.8 |
| RBCs | 14.3% | 2.6% | 0.95 |

## Acknowledgements

We thank *Helmstaedter et al. (2013)* for making their data available. This work was funded by the DFG (EXC 307 and BE 5601/1–1) and the BMBF through the BCCN Tübingen (FKZ 01GQ1002) and the Bernstein Award to PB (FKZ 01GQ1601).

## Additional information

### Funding

| Funder | Grant reference number | Author |
| --- | --- | --- |
| Deutsche Forschungsge-meinschaft | EXC 307 | Thomas Euler |
| Bundesministerium für Bildung und Forschung | FKZ 01GQ1601 | Philipp Berens |
| Deutsche Forschungsge-meinschaft | BE 5601/1-1 | Philipp Berens |
| Bundesministerium für Bildung und Forschung | FKZ 01GQ1002 | Thomas Euler |

The funders had no role in study design, data collection and interpretation, or the decision to submit the work for publication.

### Author contributions

CB, Anatomical tracing, Analysis and interpretation of data, Drafting or revising the article; TS, Anatomical tracing, Conception and design, Analysis and interpretation of data, Drafting or revising the article; SH, TE, Conception and design, Drafting or revising the article; PB, Conception and design, Analysis and interpretation of data, Drafting or revising the article

### Author ORCIDs

Thomas Euler, http://orcid.org/0000-0002-4567-6966
Philipp Berens, http://orcid.org/0000-0002-0199-4727

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
