## [Decision Letter]

Thank you for submitting your article "Connectivity map of bipolar cells and photoreceptors in the mouse retina" for consideration by *eLife*. Your article has been favorably evaluated by David Van Essen (Senior Editor) and three reviewers, one of whom, Fred Rieke (Reviewer #1), is a member of our Board of Reviewing Editors.

The reviewers have discussed the reviews with one another and the Reviewing Editor has drafted this decision to help you prepare a revised submission.

We all agreed that the paper provides important and interesting data on connectivity in the outer retina, and that the results were generally well presented. We all also agreed on two key points that need to be strengthened in the paper:

1) Separation of false vs. true contacts. The EM dataset used in the paper does not permit unambiguous identification of synapses, and hence the authors rely on an automated procedure to separate true synaptic contacts from other membrane appositions between two cells. This procedure is verified by comparison with results from human annotation. More details of this procedure and its validity are needed. Some specific items that would help are: (1) examples of contacts misidentified by the algorithm and corrected by human observers; (2) a discussion from the beginning of the results of the impact of errors in identification for the conclusions of the paper; and, (3) some discussion of the fact that there is no unambiguous "ground truth" identification available (i.e. that human observers also likely make some errors) and how that could impact the results.

2) Discussion of novel contacts. A strength of the paper is quantification of unexpected synaptic contacts. We felt it would help readers interpret the importance of those results if they were put in the context of physiological signaling (both established and predicted), and in the context of development (e.g. perhaps these contacts play a role in development rather than signaling in the mature retina – see comments from reviewer 2 in particular).

These points and other points are described in more detail in the individual reviews below.

*Reviewer #1:*

This paper used an existing electron microscopy data set to provide a comprehensive analysis of connectivity in the outer retina in mouse. The results provide direct evidence for several unexpected connectivity features that had been suggested previously but not proven definitively; further, the paper shows that a newly discovered cone bipolar type exhibits properties very distinct from the others. These results are quite interesting and are generally presented clearly. I have a few concerns about the analysis and some suggestions about presentation:

1) Identification of S cones.

S cones were identified based on the assumption that they alone were contacted by type 9 bipolar cells at invaginating contacts made at the end of the bipolar dendrites. These assumptions result in an approximately correct number of S cones given past estimates of their density. Nonetheless, the process seems almost certain to misidentify some cones. It is not entirely clear how important this is for the conclusions of the paper. The section in the Discussion presenting an alternative identification of S cones is quite helpful in this regard, but in my view, comes too late. I would like to see a more thorough analysis of the potential errors in identification and the impact of those errors on the conclusions of the paper. Doing that from the start should help build a reader's confidence. Related to this point, it would help to lay out the criteria very directly (e.g. I was not entirely clear from the second paragraph of the subsection “Identification of S- and M-cones” if contacts only at the end of the dendrites was a hard criterion).

2) Identification of synaptic contacts.

The original dataset does not permit clear identification of synapses, so the raw data the authors have to work with consists of points of close apposition of two cells. Synapses were identified automatically based on extracted features of each close apposition. These identifications were compared with those from manual identification to verify the accuracy of the automated process. At present, it is hard for a reader to evaluate and confirm the accuracy of identification of true contacts vs. false contacts, since this relies on a "ground truth" that itself has some uncertainty at least for the reader. I think it is important to construct as strong an argument as possible that manual inspection and classification indeed provides nearly unambiguous identification of true synapses. As for the point above, I would suggest doing this from the start of the paper. It would be very helpful to have a better sense for the impact of classification errors (which are not so rare from the analysis described in the last paragraph of the subsection “Classification of photoreceptor-BC contacts”) on the conclusions. Related to this point is the possibility of distant contacts or near contacts detecting diffusing transmitter. This is brought up late in the paper (subsection “Sparse contacts between some ON CBC types and cones”, last paragraph), but it would help to deal with the issue earlier (e.g. around subsection “Classification of photoreceptor-BC contacts”, first paragraph). Another point related to this is the uniqueness of identification of basal contacts – e.g. how certain can we be that rod-Off CB basal contacts are real synapses?

*Reviewer #2:*

On the whole this is a fine analysis of photoreceptor – bipolar cell contacts in the mouse retina based on an existing medium resolution SBFSEM connectome. The key new conclusions are that many rod BCs actually contact a few cones (though I think the significance is developmental rather than physiological) and many OFF cone BCs contact a few or several rods based on subtype.

Despite the impoverished resolution inherent in SEM datasets (no ribbons, vesicles, gap junctions etc.), it is possible to use proximity to suggest possible connectivity. From my experience, there will be few systematic errors in this process, unlike the inner plexiform layer where proximity can rarely be used, if ever.

The findings emphasize some new bipolar cell classes and will be very useful in guiding higher resolution TEM connectomics in process in other labs.

On the whole it is a nicely illustrated paper.

*Reviewer #3:*

In this elegant and well-executed study, Behrens and colleagues perform a detailed volume reconstruction of previously available and published serial EM data from the mouse retina. The original reconstruction by Helmstaedter and colleagues identified a new bipolar cell type (the CBCX) and focused on reconstructing the inner plexiform layer. The authors of the current study perform a reconstruction of the synaptic connections in the outer plexiform layer. This study offers significant novel findings that add to and also challenge the existing knowledge of outer retina connectivity. Most importantly, they show that there is a potential cross-over of input from cones to rod bipolar cells with the majority of RBCs (~75%) contacting cones. This further challenges the status quo of separate and isolated channels for parallel processing in the vertebrate retina. Furthermore, the authors show that the newly discovered CBCX makes surprisingly few contacts with the cones (~20%) available within the cell's dendritic reach. The work also challenges the assumption that there is equal divergence between one cone and all cone bipolar cell types.

An issue of concern, as the authors themselves point out, is that the tissue is devoid of typical synaptic markers like presynaptic ribbons, synaptic vesicles, and postsynaptic densities, which introduces a challenge when trying to decide whether contacts between pre and postsynaptic cells represent a functioning synapse. The authors' statements about connectivity should be qualified for this reason. The authors describe "true" and "false" synapses, but these judgements are unclear given that the dataset did not contain synaptic information that could be used to verify contacts. Furthermore, the authors did not illustrate "false" contacts, making the interpretation of the "true" contacts ambiguous. To this point, the subsection “Does a ‘contact’ represent a synaptic connection?” of the Discussion mentions contacts and synaptic connections and "errors" in classifying a contact. Can the authors show examples in a supplementary figure of what these human and algorithm errors look like? Such clarification would be useful for establishing common guidelines for contact classification in EM data of the retina.

---

## [Author Response]

*We all agreed that the paper provides important and interesting data on connectivity in the outer retina, and that the results were generally well presented. We all also agreed on two key points that need to be strengthened in the paper:*

*1) Separation of false vs. true contacts. The EM dataset used in the paper does not permit unambiguous identification of synapses, and hence the authors rely on an automated procedure to separate true synaptic contacts from other membrane appositions between two cells. This procedure is verified by comparison with results from human annotation. More details of this procedure and its validity are needed. Some specific items that would help are: (1) examples of contacts misidentified by the algorithm and corrected by human observers; (2) a discussion from the beginning of the results of the impact of errors in identification for the conclusions of the paper; and, (3) some discussion of the fact that there is no unambiguous "ground truth" identification available (i.e. that human observers also likely make some errors) and how that could impact the results.*

We thank the reviewers for bringing up these points. We agree that this is a limitation of the dataset we used, as it does not include specific synaptic markers.

Regarding point 1, we prepared a figure showing examples of where human and automatic labels disagree (now Figure 2—figure supplement 2). In addition, we verified that our procedures are equally likely to make mistakes for all cell types, a fact we now mention in the text.

Regarding point 2, we improved our Discussion of the contact identification procedure to discuss the possible impact of errors in contact identification. The error across all contacts is about 10%, and this error is uniform across BC types. Therefore, classification errors should not systematically distort the connectivity diagram. In addition, we followed the suggestion by reviewer 1 and moved the section describing the alternative s-cone classification further to the beginning of the paper.

Regarding point 3, we think that the highly stereotypical anatomical organization of the photoreceptor/BC synapse allows human observers to provide labels that are very close to ground truth (see comment by reviewer 1 below). We now emphasize this point in the paper.

Taken together, we are confident that the central results of our paper – no further M-/S-cone selective BCs; unconventional contacts in CBC X; rod/cone-BC contact patterns – do not depend on errors of our contact identification procedure.

*2) Discussion of novel contacts. A strength of the paper is quantification of unexpected synaptic contacts. We felt it would help readers interpret the importance of those results if they were put in the context of physiological signaling (both established and predicted), and in the context of development (e.g. perhaps these contacts play a role in development rather than signaling in the mature retina – see comments from reviewer 2 in particular).*

This is a good suggestion. We added additional text to the Discussion summarizing our understanding of how in particular the unexpected contacts fit into the current picture, i.e. with respect to physiology and development.

*These points and other points are described in more detail in the individual reviews below.*

*Reviewer #1:*

*This paper used an existing electron microscopy data set to provide a comprehensive analysis of connectivity in the outer retina in mouse. The results provide direct evidence for several unexpected connectivity features that had been suggested previously but not proven definitively; further, the paper shows that a newly discovered cone bipolar type exhibits properties very distinct from the others. These results are quite interesting and are generally presented clearly. I have a few concerns about the analysis and some suggestions about presentation:*

1) Identification of S cones.

*S cones were identified based on the assumption that they alone were contacted by type 9 bipolar cells at invaginating contacts made at the end of the bipolar dendrites. These assumptions result in an approximately correct number of S cones given past estimates of their density. Nonetheless, the process seems almost certain to misidentify some cones. It is not entirely clear how important this is for the conclusions of the paper. The section in the Discussion presenting an alternative identification of S cones is quite helpful in this regard, but in my view, comes too late. I would like to see a more thorough analysis of the potential errors in identification and the impact of those errors on the conclusions of the paper. Doing that from the start should help build a reader's confidence. Related to this point, it would help to lay out the criteria very directly (e.g. I was not entirely clear from the second paragraph of the subsection “Identification of S- and M-cones” if contacts only at the end of the dendrites was a hard criterion).*

We thank the reviewer for these suggestions. We improved our description of the criteria for identifying S-cones and moved the section with the alternative analysis from the Discussion to the Results. Note that the two analyses represent the two extremes of the spectrum for S- cone identification and both suggest that there are no chromatic channels in addition to CBC1 and CBC9.

2) Identification of synaptic contacts.

The original dataset does not permit clear identification of synapses, so the raw data the authors have to work with consists of points of close apposition of two cells. Synapses were identified automatically based on extracted features of each close apposition. These identifications were compared with those from manual identification to verify the accuracy of the automated process. At present, it is hard for a reader to evaluate and confirm the accuracy of identification of true contacts vs. false contacts, since this relies on a "ground truth" that itself has some uncertainty at least for the reader. I think it is important to construct as strong an argument as possible that manual inspection and classification indeed provides nearly unambiguous identification of true synapses. As for the point above, I would suggest doing this from the start of the paper.

We thank the reviewer for pointing this out. We think that the highly stereotypical anatomical organization of the photoreceptor/BC synapse allows nearly unambiguous identification of contacts by a trained human observer (see also our reply to the overall assessment). We now explain and emphasize this point already in the beginning of the paper.

*It would be very helpful to have a better sense for the impact of classification errors (which are not so rare from the analysis described in the last paragraph of the subsection “Classification of photoreceptor-BC contacts”) on the conclusions.*

Classification errors (that is, where the algorithm did not agree with the human observer) were nearly evenly distributed across BC types, indicating that the connectivity profiles are not distorted by these errors. We now mention this in the Results and the Discussion (see also our reply to the overall assessment).

*Related to this point is the possibility of distant contacts or near contacts detecting diffusing transmitter. This is brought up late in the paper (subsection “Sparse contacts between some ON CBC types and cones”, last paragraph), but it would help to deal with the issue earlier (e.g. around subsection “Classification of photoreceptor-BC contacts”, first paragraph).*

We thank the reviewer for this suggestion. However, as reviewer #2 is highly critical of this possibility we refrained from giving this point more weight by mentioning it earlier in the paper. We changed the Discussion to reflect reviewer #2’s criticism.

*Another point related to this is the uniqueness of identification of basal contacts – e.g. how certain can we be that rod-Off CB basal contacts are real synapses?*

We think there is good evidence for this. For example, Hack, Peichl and Brandstätter (PNAS 1999) have shown that AMPA-type glutamate receptors are expressed at the basal contacts between rods and OFF CBCs, making it likely that these basal contacts are active synaptic sites. We now cite this in the Discussion.

*Reviewer #3:*

[…]

*An issue of concern, as the authors themselves point out, is that the tissue is devoid of typical synaptic markers like presynaptic ribbons, synaptic vesicles, and postsynaptic densities, which introduces a challenge when trying to decide whether contacts between pre and postsynaptic cells represent a functioning synapse. The authors' statements about connectivity should be qualified for this reason. The authors describe "true" and "false" synapses, but these judgements are unclear given that the dataset did not contain synaptic information that could be used to verify contacts. Furthermore, the authors did not illustrate "false" contacts, making the interpretation of the "true" contacts ambiguous. To this point, the subsection “Does a ‘contact’ represent a synaptic connection?” of the Discussion mentions contacts and synaptic connections and "errors" in classifying a contact. Can the authors show examples in a supplementary figure of what these human and algorithm errors look like? Such clarification would be useful for establishing common guidelines for contact classification in EM data of the retina.*

We added examples of contact points where our algorithm did not agree with the human labels (Figure 2—figure supplement 2).